# Genome-Wide Analysis and Functional Characterization of *LACS* Gene Family Associated with Lipid Synthesis in Cotton (*Gossypium* spp.)

**DOI:** 10.3390/ijms24108530

**Published:** 2023-05-10

**Authors:** Yike Zhong, Yongbo Wang, Pengtao Li, Wankui Gong, Xiaoyu Wang, Haoliang Yan, Qun Ge, Aiying Liu, Yuzhen Shi, Haihong Shang, Yuanming Zhang, Juwu Gong, Youlu Yuan

**Affiliations:** 1College of Plant Science and Technology, Huazhong Agricultural University, Wuhan 430070, China; xiaozhongyk@126.com (Y.Z.); soyzhang@mail.hzau.edu.cn (Y.Z.); 2State Key Laboratory of Cotton Biology, Institute of Cotton Research, Chinese Academy of Agricultural Sciences, Anyang 455000, China; lipengtao1056@126.com (P.L.); gongwankui@caas.cn (W.G.); wangxiaoyu67@126.com (X.W.); yanhl1989@163.com (H.Y.); gequn@caas.cn (Q.G.); liuaiying@caas.cn (A.L.); shiyuzhen@caas.cn (Y.S.); shanghaihong@caas.cn (H.S.); 3Cotton Sciences Research Institute of Hunan, National Hybrid Cotton Research Promotion Center, Changde 415101, China; wangyongbo040923@163.com

**Keywords:** cotton, long-chain acyl-coenzyme A (CoA) synthetase (*LACS*), fatty acids, expression analysis

## Abstract

Cotton *(Gossypium* spp.) is the fifth largest oil crop in the world, and cottonseed provides abundant vegetable oil resources and industrial bioenergy fuels for people; therefore, it is of practical significance to increase the oil content of cotton seeds for improving the oil yield and economic benefits of planting cotton. Long-chain acyl-coenzyme A (CoA) synthetase (*LACS*) capable of catalyzing the formation of acyl-CoAs from free fatty acids has been proven to significantly participate in lipid metabolism, of which whole-genome identification and functional characterization of the gene family have not yet been comprehensively analyzed in cotton. In this study, a total of sixty-five *LACS* genes were confirmed in two diploid and two tetraploid *Gossypium* species, which were divided into six subgroups based on phylogenetic relationships with twenty-one other plants. An analysis of protein motif and genomic organizations displayed structural and functional conservation within the same group but diverged among the different group. Gene duplication relationship analysis illustrates the *LACS* gene family in large scale expansion through WGDs/segmental duplications. The overall Ka/Ks ratio indicated the intense purifying selection of *LACS* genes in four cotton species during evolution. The *LACS* genes promoter elements contain numerous light response cis-elements associated with fatty acids synthesis and catabolism. In addition, the expression of almost all *GhLACS* genes in high seed oil were higher compared to those in low seed oil. We proposed *LACS* gene models and shed light on their functional roles in lipid metabolism, demonstrating their engineering potential for modulating TAG synthesis in cotton, and the genetic engineering of cottonseed oil provides a theoretical basis.

## 1. Introduction

Cotton is the fifth largest oil-bearing crop in the world, and cottonseed provides abundant oil resources. At present, most of the existing studies focus on fiber yield and quality traits, while relatively few research works on cottonseed quality have been reported, which leads to a shortage of varieties with high fiber quality and high cottonseed oil content, resulting in the low utilization efficiency of cottonseed’s additional value [1]. Cottonseed oil has been receiving increasing attention on the grounds of the increasing worldwide demand for vegetable oil. Regarding the nutritional value of cottonseed oil, it provides essential amino acids and contains vitamin E, free gossypol, neuroactive N-acylethanolamines, and phytosterols and unsaturated fatty acids polyunsaturated fatty acids, which account for 71% of the seed oil, of which more than 50% belongs to linoleic acid (C18:2). It is also rich in palmitic acid (C16:0) and oleic acid (C18:1). The utilization of cottonseed oil as food could be categorized into cooking, marinades, pastries, dressing, margarines, etc. [2]. Besides being edible, cottonseed oil can also be used as raw materials for soap, pesticide reagents, lubricating oil [3], and biodiesel [4] in industry.

Long-chain acyl-CoA synthetases (*LACS*) are a subgroup of the carboxyl-CoA ligase superfamily, also known as acyl-activating [5,6], and are prominent in fatty acid (FA) synthesis and catabolism. Similar to glucose, FA must be activated before it can be metabolized in cells, and the enzymes that catalyze FA activation are called fatty acyl-CoA synthetases. The catalytic reactions can be divided into two main steps: in the first step, FA combines with ATP and react to form a fatty acid to form an enzyme-bound acyl-adenylate intermediate and pyrophosphate. Subsequently, this intermediate is attacked by a CoA to produce an acyl-CoA and an adenosine monophosphate (AMP). Fatty Acid + ATP—Mg^2+^ → acy-AMP + PPi, acy-AMP + CoA-SH—Mg^2+^ → acy-S-CoA + AMP [7]. These enzymes play a vital role in lipid synthesis and storage, fatty acid catabolism, vectorial acylation, and the synthesis of cutin polyesters and cuticular waxes [8]. Previous research works have shown that *LACSs* are involved in intracellular FA homeostasis [9] and FA transport in bacteria [10], yeast [11], and mammals [12].

In higher plants, *LACSs* are involved in quite a few pathways and localized in different cellular organelles. As a model organism of plants, the functions of nine *AtLACS* genes in *Arabidopsis* have been discovered and confirmed successively. *AtLACS1*, a major isoform involved in the synthesis of wax, activates very long chain fatty acids (VLCFAs) (20:0–30:0) with the highest specificity for triacontanoic acid (30:0) and higher activity with palmitic. In accord with this, mutants with a loss of function of *AtLACS1* have substantial reductions in 16:0-derived cutin monomers [5]. *AtLACS2* takes charge of cutin synthesis and prefers hydroxylated over unsubstituted FAs, with enhanced activity toward hydroxylated 16:0 [13]. Previous studies of *AtLACS2* mutation observed strong cutin deficiency with few effects on wax biosynthesis under normal growth conditions [5,13]. Ian P. Pulsifer [14] pointed out that *AtLACS3* was involved in the trafficking of intracellular lipids towards cuticle biosynthesis through the transmembrane movement of fatty acids. The results of the experiment indicated that a yeast *fat1D* mutant was deficient in both very-long-chain acyl-CoA synthetase activity and exogenous fatty acid uptake. Additionally, the heterologous expression of *AtLACS3* is able to complement both of these deficiencies [14]. It was reported that *AtLACS4* and *AtLACS9* jointly participate in the process of lipid flux from the endoplasmic reticulum (ER) to the plastid for glycerolipid synthesis, but their catalytic activity is lower than that of *AtLACS1* and *AtLACS2* [15,16] Thus, *AtLACS4* and *AtLACS9* appear to play redundant roles in wax and cutin synthesis. Studies on *AtLACS6* and *AtLACS7* mutants showed that the lipid metabolism pathway was blocked, and seeds could only germinate through exogenous sucrose [15]. To some extent, *AtLACS6* and *AtLACS7* have similar functions, and both could catalyze most fatty acids. *AtLACS8* and *AtLACS9* seem to play the same role as *AtLACS4* [15,16,17]. The overall function of *AtLACS5* has not been revealed till now. In short, most of the *AtLACS* genes are expressed in flowers and germinated seeds, which indicates that these genes not only participate in the lipid metabolism of flower tissues, but also play a key role in the synthesis of seed glycerides.

The *LACS* genes are characterized not only in *Arabidopsis*, but also in other plants. For example, *GhACS1* in cotton shows a high degree of homology with *AtLACS4* and *AtLACS5*, is required for normal microsporogenesis. In vitro enzyme activity analysis revealed that *GhACS1* prefers long-chain FAs with the highest activity toward oleic acid [18]. *GmACSL2* is a peroxisomal isoform of soybean *LACS*, which promotes the degradation of fatty acids and lipids during seed germination [19]. *HaLACS1* and *HaLACS2* are highly expressed in developing seeds and perform essential roles in the oil synthesis of sunflower seeds, which displays sequence homology with *AtLACS9* and *AtLACS8* genes, respectively [20]. *LuLACS8* identified from flax shows substrate preference towards α-linoleic acid and contributes to the biosynthesis of α-linolenic acid [21]. The *MdLACS2* gene from apple is involved in wax biosynthesis in apple [22]. *BnLACS2* from rapeseed, an orthologue of the *AtLACS2* gene, is predominantly expressed in developing seeds and involved in seed oil production [23]. The above studies demonstrated that *LACS* enzymes play conserved roles in higher plants.

In this study, bioinformatics methods were used to conduct whole-genome identification, evolutionary analysis on the *LACS* genes family of cotton and to unravel the gene structure features, chromosomal locations, phylogenetic relationships, synteny, and expression patterns to highlight the potential functional diversity. Our results will provide useful theoretical support for the functional characterization of the *LACS* genes that are involved in the fatty acid synthesis process in cotton.

## 2. Results

### 2.1. Identification of LACS Genes

We submitted the *LACS* protein sequences of *Arabidopsis* to the Pfam database, finding that these genes contain an AMP conserved domain. The AMP domains play a major part in ATP binding and enzyme catalysis [24]. Then, the HMM file of the AMP conserved domain was downloaded, and the *Gossypium hirsutum*, *Gossypium arboreum*, *Gossypium raimondii*, and *Gossypium barbadense* reference genomes were screened to obtain the potential gene containing the AMP domain. In total, 22, 20, 12, and 11 *LACS* genes were identified in *Gossypium hirsutum*, *Gossypium barbadense*, *Gossypium raimondii*, and *Gossypium arboreum*, respectively. The number of *LACSs* in tetraploid cotton (*Gossypium hirsutum* and *Gossypium barbadense*) is almost twice that of diploid cotton (*Gossypium raimondii* and *Gossypium arboreum*), which reconfirms that the allotetraploid of cotton is formed by doubling the chromosomes after the hybridization between A-genome diploid and D-genome diploid [25].

Subsequently, the biophysical properties of the *LACS* genes in four cottons including genomic position, coding sequence length, molecular weight, protein length, isoelectric point (pI), electric charge, exon-intron and subcellular localization prediction-related information were determined (Appendix A). All of *LACS* genes identified above were named based on the order of their chromosomal location. The sequence analysis revealed that these four cotton LACS proteins did not show remarkable difference in length, which mainly ranged from 618aa to 731aa. The predicted molecular weights of the 65 LACS proteins mainly ranged from 70.884 to 80.48, and pI values ranged from 5.662 to 10. The prediction of subcellular localization shows that there are 26 genes located in the chloroplast, 22 in the cytoplasm, 14 in the nuclear, 2 in the extracellular matrix and 1 in the mitochondria. As expected, this result may be because the site of fatty acid biosynthesis in the seeds is mainly in the plastid, and the endoplasmic reticulum is the site where the fatty acids and glycerides synthesized in the plastid are assembled into triacylglycerols.

### 2.2. Phylogenetic Analysis of LACS Genes Family in Cotton

The sequences of the 22 GhLACS, 20 GbLACS, 12 GrLACS, and 11 GaLACS proteins and LACS proteins in other species were analyzed with ClustalW software (Version 2.1) using default parameters. The LACS proteins were assigned to six groups in higher plants, such as groups A-G (Figure 1 and Appendix A) using MEGA11 software (Version 11.0.11) [26] via the neighbor-joining method with the following settings: bootstrap method, 1000 replicates; Poisson model; and pairwise deletion. The phylogenetic tree consisted of several species, ranging from At, *Arabidopsis thaliana*; Gh, *Gossypium hirsutum*, Gb, *Gossypium barbadense*; Gr, *Gossypium raimondii*; Ga, *Gossypium arboretum*; Cs, *Camelina sativa*; Bn, *Brassica napus*; Si, *Sesamum indicum*; Oe, *Olea europaea*; Ha, *Helianthus annuus*; Gm, *Glycine max*; Ah, *Arachis hypogaea*; Bn, *Brassica napus*; Cs, *Camelina sativa*; Tur, *Triticum Urartu*; Zm, *Zea may*; Aco, *Ananas comosus*; Eg, *Elaeis guineensis*; Mac, *Musa acuminate*; Nta, *Nicotiana tabacum*; Osa, *Oryza sativa*; Smo, *Selaginella moellendorffii*; Ppa, *Physcomitrella paten*; Esi, *Ectocarpus siliculosus*; Gsu, *Galdieria sulphuraria.*

Groups A and B include *LACS1* and *LACS2,* respectively, and some hypothetical sequences. Group C consists of *LACS3*-*5*, while *LACS6* and *7* were dominant in Group D. Subsequently, Group E includes *LACS8*, and Group F contains *LACS9*. (Appendix A, Figure 1). However, we observed that the LACS proteins in lower plant algae (Esi and Gsu) did not have a grouping. In addition, we found the LACS4 proteins in *Selaginella moellendorffii* and *Physcomitrella patens* were not assigned to Group C. In all groups, algal (Esi and Gsu), bryophytes (Ppa) and fern (Smo) were phylogenetically divergent from terrestrial angiosperms.

Within Group A, Gm, Ah, Gh, Gb, Ha, Cs and Eg contain more than one sequence. Bn, Cs, Gh, Gm, and Ah in group B contain more than one sequence while in Group C, Group D, Group E and Group F, 14, 14, 6 and 15 species contain more than one sequence, respectively. More than one sequence in many Groups possibly present gene duplication events. Group D was the largest clade consisting of 76 sequences of *LACS6*-*7* of all angiosperms. Group F was the second largest group consisting of 67 sequences of *LACS9*. Overall, the phylogenetic relationship of *LACS* genes from all species reveals that *LACS* genes are broadly distributed in flowering plants, and the copy number was expanded in most species.

Previous published studies showed that acyl-activating enzymes contain two highly conserved domains [27,28,29], Motif 1 corresponds to aa “(T/S) (S/G) G (T/S) (T/E) GNPKG” and contains a putative AMP binding domain. Motif 2 contains 36–37 aa and the conserved arginine (R), corresponding to aa” TGDxxxxxxxGxxxhx[DG]RxxxxhxxxxGxxhxx[EK]hE”. The x in the sequence indicates any aa, and h indicates a hydrophobic residue. The multiple sequences alignment of the GhLACS, GaLACS, GbLACS, GrLACS and AtLACS proteins showed that almost all LACS proteins contained these two conserved motif domains (Figure 2).

### 2.3. Organization of LACS Genes on Chromosomes of Four Cotton Species

Chromosomal maps of *LACS* genes from two tetraploid ((AD)1, (AD)2) and two diploid ((AA), (DD)) cotton species were constructed to further examine their structural and evolutionary dynamics associated with genomic distribution. A total of 64 *LACSs* were assigned to their specific chromosomes, while the rest of one *LACS* was awarded to unmapped scaffolds signifying the higher degree of maturation in AMP-binding superfamily evolution.

Among 22 *GhLACSs*, 11 were located on 7 chromosomes of At sub-genome (GhAt) and 11 *LACSs* were positioned on 7 chromosomes at Dt sub-genome (GhDt). Overall, to some extent, a uniform distribution pattern of *GhLACS* was noted in both sub-genomes GhAt and GhDt except chromosome 2 and 3. The rest of the chromosomes possess almost a similar number of *LACS* members. Furthermore, the number of genes located on the chromosomes of GhDt sub-genomes compared with the chromosome of their parent orthologues D/D genomes vary to a little extent, which implied that the *GhLACS* family might have lost or gained various genes during evolution. For *G. barbadense*, 20 *GhLACSs* were mapped on 14 chromosomes, of which At sub-genome (GbAt) harbors 10 genes and Dt sub-genome (GbDt) contains 10 *LACSs*. Interestingly, the distribution of *GhLACS* and *GbLACS* on At and Dt sub-genomes were highly similar. The comparative analysis of GhAt/Dt and GbAt/Dt sub-genomes with their diploid progenitors (A and D Genomes) demonstrates surprising results. Both of GhAt and GbAt sub-genome showed higher conservation for an even distribution of *LACS* members with its diploid progenitor *G. arboretum*. However, there is an uneven distribution among the orthologues of Dt sub-genome and its progenitor *G. raimondii* and *G. hirsutum*, indicating the possibility of gene loss during evolution (Figure 3, Table 1).

The 11 *GaLACSs* and 12 *GrLACSs* were mapped on seven chromosomes of A-Genome and eight chromosomes of D-Genome, respectively, which were unevenly distributed on chromosomes of both species and one *GaLACS* could not be anchored onto any specific chromosome (Figure 3, Table 1). The overall comparative analysis of diploid pro-genitors and their originators revealed that the *LACS* genes were evenly distributed among orthologue and paralogue chromosomes, and they also faced several exceptions of uneven distribution representing the degree of gene loss during evolution which exhibited the contribution of diploid progenitors in tetraploid cotton species and its expansion during evolution.

### 2.4. Gene Duplication Relationship and Collinearity Analysis of LACS Genes

Gene duplications are considered to be one of the primary driving forces in the evolution of genomes and genetic systems [30]. Duplicated genes provide raw material for the generation of new genes, which, in turn, facilitate the generation of new functions. Segmental duplication, tandem duplication, and transposition events, such as retroposition and replicative transposition [31] are considered to represent three principal evolutionary patterns [32,33,34]. To elucidate the expanded mechanism of the *LACS* genes family, we performed a gene duplication event analysis including tandem duplication and WGDs/segmental duplications in the four cotton species (two diploid progenitors (AA) and (DD) and two tetraploid descendants (AD)1 and (AD)2).

The expansion pattern of *LACS* genes were studied using circos analysis in four species of cotton, which revealed the duplication of various genes. Paralogous pairs of *LACS* genes family in the four cotton species were arising from WGDs/segmental duplications, and no tandem duplications were found. In *G. arboreum*, seven singletons and three WGDs/segmental duplications were identified in *LACS* genes. Similarly, seven and four genes exhibited singletons and WGDs/segmental duplications in *G. raimondii*. In tetraploid cotton species, *G. hirsutum* showed 3 singleton and 30 WGDs/segmental gene duplications whilst 2 singletons and 29 segmental duplications were observed in *G. barbadense* (Figure 4, Appendix A). Of interest, in addition to 66 segmentally duplicated genes in our study, there are 19 singletons. This finding indicates that the remaining 19 segmentally duplicated *LACS* genes might be derived from independent duplication events. Therefore, our findings indicate that most of genes involved in segmental duplication are a result of WGDs/segmental events, while the remainder may have arisen as a result of independent duplication events.

Further, results of homologous blast exhibited the collinearity between *G. arboreum*, *G. raimondii*, *G. barbadense* and *G. hirsutum*. Meanwhile, compared with A-genome, D-genome comprises a greater number of collinear genes with AD1 and AD2 genomes. According to MCScanX analysis, the A-genome exhibited 32 duplicated gene pairs with *G. hirsutum* and *G. barbadense*. Similarly, Diploid D-genome was found containing 40 duplicated gene pairs for each tetraploid species *(G. hirsutum* and *G*. *barbadense)* (Figure 5, Appendix A). Overall, these duplicated gene pairs of four cotton species show the basis for polyploidization and large-scale expansion of *LACS* genes family during evolution.

The non-synonymous substitution rate (Ka), synonymous substitution rate (Ks) and KA/KS of 65 pairs of repetitive sequences were calculated to reveal the selection pressure of *LACS* genes family in the process of evolution (Figure 5B, Appendix A). The Ka/Ks of duplicated gene pairs were all less than 1, which tended to be pure selection, indicating that *LACS* genes sequence similarity was very high and relatively conserved during evolution. The evolution time of the duplicated events of *LACS* genes can be divided into three evolution periods (Figure 5C, Appendix A). The first period was 22.31–34.93 million years ago (Mya), with nine duplicated gene pairs. A total of 38 *LACSs* duplicated gene pairs occurred at 11.85–17.61. The remaining 18 *LACSs* duplicated gene pairs occurred at 0.46–1.99 Mya. Notwithstanding, while these homology gene sequences were conserved, they were different in evolutionary time.

### 2.5. Gene Structure and Conserved Motifs of LACSs

We examined the gene structure, motif, and protein domains of the *LACS* genes identified in four cottons and *Arabidopsis* (Figure 6). The exon/intron structures were then compared with conserved protein motifs subgroups, which showed that the *LACS* genes family all have exons and introns, and genes in the same subgroup usually have similar exon/intron structures. Except for the intron sequences of *GbLACS7*.1, *GhLACS7*.1, *GrLACS7*.1, and *GhLACS7*.5 are longer than those in the same subgroup. The analysis revealed that the distribution of exon regions in *LACS* varies from 11 to 23 in numbers. Nonetheless, it’s worth mentioning that tightly clustered *LACS* genes were more analogous in arrangements and had the same number of exons but varied in exon/intron lengths (Figure 6C).

To analyze the conserved motif of LACS protein, we used MEME online software (Version 5.4.1) to analyze the 74 protein sequences. In total, 10 motifs were identified by the online program MEME and named Motif 1–10 (Figure 6B and Appendix A). Motifs 1,2, 4, 6, 8, 7 and 9 as shown in Figure 6B were common in all LACS proteins, indicating that these proteins were highly conserved. The analysis displayed that the same subgroup had similar motifs, while different subgroups had differences, indicating that the genes had evolved some specific functions while being conserved. In accordance with InterProScan and GenomeNet annotations, Motifs 1, 3, 4 and 5 were an AMP-binding enzyme domain. Some specific amino acids residues within the motifs were associated with AMP-binding signatures, as shown in previous studies, for example, motifs 1 and 4 [28]. These motifs have a prominent function in LACS proteins. All of these recognized motifs are part of an AMP-dependent synthetase that plays an important role in lipid biosynthesis and intracellular distribution, which is in line with previous studies [35].

### 2.6. Orthologous Gene Clusters Identification

The relative assessment was ascertained to detect orthologous *LACS* genes clusters in four cottons, which would assist in evaluating polyploidization events in the evolutionary process of *LACS* genes family in cotton A and D genomes. As can be seen from Figure 7A, *G. hirsutum*, *G. arboreum* and *G. raimondii* all have six gene clusters, and *G. barbadense* has five gene clusters. Results also revealed the common presence of five orthologous gene clusters which are solely composed of genes found in cotton species indicating that polyploidization has resulted in the evolution of new cotton specific orthologues gene clusters. Further analysis was carried out with GO annotation, finding that cluster 2 (GO:0006631), cluster 3 (GO:0006631), and cluster 4 (GO:0010025)-enriched 2 GO terms related to lipid metabolism (fatty acid metabolic process and wax biosynthetic process).

### 2.7. Cis-Element Analysis in the Promoter Regions of the LACS Genes in Cotton

Cis-elements (CEs) are transcription initiation regions of particular genes that regulate gene expression at various developmental stages by controlling the transcription of nearby genes. CEs sequences, such as enhancers, silencer, insulators and promoters, control development and physiology by regulating gene expression. A dvergence in CREs is the major driving force for evolution causing functional advances [36,37]. From the analysis above, key findings include a significant amount of cis-elements involved in various cellular processes. In this work, the different subgroups are slightly different in function, but the overall functions are similar. Particularly, the elements related to light response (Box, 32.15%) (Figure 8A,B) are abundant in four cotton, and light response (GT1-motif, 17.48%), growth and development (CAT-box, 30.51% and TGA-element, 20.34%), hormone responsive (ABRE, 38.03%), stress responsive (MBS,42.61%) are also ubiquitous. By the above analysis, we found that the cis-elements related to light response are abundant in every gene. Light is a critical environmental factor that not only drives photosynthetic carbon fixation, but also directly and indirectly regulates plant growth and development in many other ways. Tung tree seed oil accumulation is suppressed by dense shade during the rapid oil accumulation phase [38] and lipid metabolism is repressed in developing embryos grown in the dark [39], while metabolic reprogramming occurs in *Chlorella vulgaris* under high light conditions, leading to enhanced oil accumulation [40]. Thus, light influences the accumulation of lipids in embryos by inducing the expression of metabolic genes. So, the cis-elements related to light response play an important role in FA synthesis and catabolism.

### 2.8. GhLACS Genes Expression Patterns at Different Periods in Two Cotton Species

To investigate the ovule expression patterns of the *GhLACS* genes, the gene expression level of 8 selected *GhLACS* genes belonging to Group A (*GhLACS1*.2), Group B (*LACS2*.2), Group C (*LACS4*.4), Group D (*LACS7*.2, *LACS7*.4 and *LACS7*.7), Group E (*LACS8*.1) and Group F (*LACS9*.4) members were determined by using qRT-PCR in seeds during 5, 10, 20, 30 and 40 DPA stage (Figure 9) stages. Similar to the behavior observed for *Arabidopsis* [41], it can be seen from Figure 9 that the expression of almost all *LACS* genes in high seed oil cotton (High SOC, I70-177) was higher compared to those in low seed oil cotton (Low SOC37, CCRI70-037). So, *LACS* genes are tightly associated with the fatty acid synthesis in seeds.

### 2.9. A Hypothesis about the Roles of LACS in Cotton Seed Development

The phylogenetic relationship, conserved motifs and multiple sequence alignment between LACS proteins of *Arabidopsis* and cotton showed that LACS proteins belonging to the same subgroup were highly similar and conserved. Therefore, a working model was proposed for illustrating the roles of *LACS* genes in lipid metabolism in cotton based on the demonstrated functional role of *Arabidopsis* LACS proteins in different subcellular compartments [42]. Plastids are the places where de novo synthesis of FAs occurs. *AtLACS9* is known to localize to the plastid envelope. From Figure 1, Figure 2 and Figure 6, LACS9 proteins in *Arabidopsis thaliana* and cotton are highly similar. So, we think that the LACS9 proteins of cotton plays a role in the activation of de novo synthesized LCFAs in plastids (Figure 10). ER is the compartment where membrane lipids, cuticular lipids, and triacylglyceride (TAG) are produced. Most identified *AtLACSs* localize to this organelle, for example AtLACS1, AtLACS2, AtLACS4, and AtLACS8 proteins, which were also highly similar to cotton LACS proteins belonging to the same subgroup (Figure 1, Figure 2, and Figure 6). Different from the LACS proteins involved in lipid synthesis, LACS proteins mediating FA degradation usually localize to peroxisomes. AtLACS6 and *AtLACS7* proteins are found to reside in peroxisome [15,41]. On the whole, it may be inferred that *LACSs* in different subcellular compartments play different roles in lipid synthesis and degradation. We found it intriguing that *GhACS1* and its homolog *AtLACS4* did not show the same expression pattern as expected [18]. Further study of *LACSs* in cotton are required to comprehensively clarify the function of *LACS* genes during lipid metabolism.

## 3. Discussion

*LACS* genes have been identified at a genome-wide scale in various plant species, and knowledge of their potential functions in lipid synthesis and degradation in *Arabidopsis thaliana* [16], *Brassica napus* [23], *Brassica napus* [19], *Helianthus annuus* [20], *Malus domestica* [22], etc. *LACS* genes in cotton have not been identified, and few studies have been performed on their functions in cotton. By using HMM, we identified 22, 20, 12 and 11 *LACS* gene family members in *G. hirsutum*, *G. barbadense*, *G. raimondii* and *G. arboreum*, respectively. The phylogenetic tree analysis showed that cotton *LACS* genes belonged to six categories: Group A, Group B, Group C, Group D, Group F and Group G.

Through sequence alignment, it was found that all four cotton species contain two highly conserved domains, just like *Arabidopsis*. This indicates that they may have the same function and might be used as candidate genes for lipid synthesis and degradation.

Through the analysis of chromosome location and collinearity, we found that the *LACS* genes in cotton were unevenly distributed in the chromosomes. In cotton, a total of 66 orthologous/paralogous gene pairs were identified; which were predicted in WGDs/segmental duplications to form paralogous gene pairs within the GhAt/GhDt, GbAt/GbDt, A2/A2 and D5/D5 sub-genomes, furthermore 19 singleton gene. No tandem duplication gene pairs were found for *LACS* genes family.

After the analysis of the *LACS* gene’s structure and motif and protein domains, it was found that different subgroups of *LACS* genes have different numbers of introns and exons, but most of the genes had similar motifs, and recognized motifs are part of an AMP-dependent synthetase that play an important role in lipid biosynthesis.

According to orthologous gene cluster identification, elemental analysis of *LACS* genes promoters and relative expression of *GhLACS* genes. The different subgroups are slightly different in function, but the overall functions are similar and the cis-elements related to light response associated with FA synthesis and catabolism are found to be abundant in each gene The evolution of new cotton specific orthologues gene clusters owing to polyploidization. GO annotation found that almost all *LACS* gene functions are related to lipid metabolism. Furthermore, the relative expression of most *LACS* genes in high oil cotton was higher than that in low oil cotton. On the basis of the possible roles of *AtLACS* genes residing in different subcellular compartments, we proposed the roles of *LACS* genes in lipid metabolism in cotton and genetic engineering of cottonseed oil provides theoretical basis Therefore, it is possible to improve the composition and yield of cottonseed oil.

## 4. Materials and Methods

### 4.1. Databases

The whole Cotton sequence—including *Gossypium arboreum* (A2, CRI_v1), *Gossypium barbadense* (AD2, HEAU_v1), *Gossypium hirsutum* (AD1, ZJU_v2.1), and *Gossypium raimondii* (D5, JGI_v2), were downloaded from the Cottongene database (Version 1.0) (https://www.cottongen.org, accessed on 15 April 2022). The genome sequence of *Arabidopsis thaliana*, *Glycine max*, *Arachis hypogaea*, *Helianthus annuus*, *Camelina sativa*, *Brassica napus*, *Olea europaea*, *Elaeis guineensis*, *Zea mays*, *Sesamum indicum* was downloaded from the EnsemblPlants database (http://plants.ensembl.org/index.html, accessed on 17 April 2022). Other plant *LACS* sequences were derived from Zhang [29].

### 4.2. Dentification of Cotton LACS Genes Family Members

To identify cotton *LACS* candidates, the Hidden Markov Model (HMM) analysis was used for the search [43]. We downloaded HMM profile of *LACS* (PF00501) from Pfam protein family database (http://pfam.xfam.org/, accessed on 19 April 2022) and used it as the query (*p* < 0.001) to search the cotton protein sequence data. Additionally, we removed the sequences that were longer than 1000 amino acids (aa) and less than 300aa [29]. Finally, we further examine various biophysical properties of cotton *LACS* genes including protein and genomic lengths, molecular weights (MWs), isoelectric points (pIs), the number of introns and exons, and charge by using CottonFGD (https://cottonfgd.net/, accessed on 20 April 2022) [44]. We used the online website, WOLF-PSORT (https://wolfpsort.hgc.jp/, accessed on 25 April 2022) to predict the subcellular localization of LACS.

### 4.3. Phylogenetic Analysis of the LACS Proteins

The full-length amino acid sequences of Gossypium hirsutum, Gossypium arboreum, Gossypium raimondii, Gossypium barbadense, Arabidopsis thaliana, Glycine max, Arachis hypogaea, Helianthus annuus, Camelina sativa, Brassica napus, Olea europaea, Elaeis guineensis, Zea mays and Sesamum indicum etc. (Appendix A) by LACS genes were aligned with the ClustalW program (Version 2.1) with the default settings and then manually adjusted in MEGA11. Subsequently, we constructed the Maximum likelihood (ML) tree with 1000 boot-strap replicates, using the Poisson substitution model with default parameters in MEGA11 [26].

### 4.4. Chromosomal Locations and Gene Collinearity Analysis

The physical positions of chromosomal locations from four cotton species, including *Gossypium hirsutum*, *Gossypium arboreum*, *Gossypium raimondii* and *Gossypium barbadense*, were drawn with the help of TBtools software (Version 1.108) [45]. Systematic names of *LACS* genes were assigned based on their chromosome distribution. Whole proteins of the four cotton species were compared against each other using local blast software (Versions 1.2) with e value less than e^−5^. The blast outputs of all protein-coding genes were imported into MCScanX [46]. Gene duplication was assessed through MCScanX and the results were achieved by TBtools. The syntenic relationships of *LACS* genes were demonstrated in a circos map using TBtools software (Version 1.108). Duplicated gene pairs belonging from the same genome/subgenome and locating at the same chromosome with a maximum of 200 kb distance between each other are considered as Tandem duplicated [47]. Whole-genome duplications (WGDs)/segmental duplications genes through polyploidy followed by chromosome rearrangements [48]. Dispersed is defined as scattered, not classified homologous genes. Selection pressure experienced by each duplicated pair during evolution was calculated with the rate of non-synonymous (Ka) to synonymous (Ks) substitution rate with TBtools. The evolutionary duplication time (T) was calculated by T = Ks/2λ × 10^−6^ (Mya), whereas (λ) = 1.5 × 10^−8^ in cotton [49].

### 4.5. Multiple Sequences Alignment and Conserved Motifs Analyses

The multiple sequences alignment of the 22 GhLACS, 20 GbLACS, 12 GrLACS, 11 GaLACS proteins and 9 AtLACS proteins was carried out using DNAMAN software (Versions 6) with default parameters (http://dnaman.software.informer.com/, accessed on 26 April 2022). Two highly conserved domains in the *LACS* genes were discerned to further show the characteristics of the GhLACS, GbLACS, GrLACS, GaLACS proteins [27,28,29] and were exhibited via TBtools.

### 4.6. Analysis of the Conserved Protein Motifs and Gene Structure

Using amino acid sequences as an input file at the online server of MEME Suite 5.4.1 (https://meme-suite.org/meme/, accessed on 9 May 2022) [50], we detect the highly conserved motifs of GhLACS, GbLACS, GrLACS, GaLACS and AtLACS proteins. While the online server of GSDS (http://gsds.gao-lab.org/, accessed on 12 May 2022) was provided with genomic and CDS sequences of *G. hirsutum*, *G. arboreum*, *G. raimondii*, *G. barbadense,* and *Arabidopsis* encoding *LACSs* and NWK file of the phylogenetic tree to display their exon/intron organizations.

### 4.7. Sequence Based Orthologous LACS Genes Identification

With orthoVenn2 [51] (https://orthovenn2.bioinfotoolkits.net/cluster-venn, accessed on 14 May 2022), we characterized the orthologous proteins of the *G. hirsutum*, *G. arboreum*, *G. raimondii*, and *G. barbadense*. Protein sequences of identified *LACS* genes from all four cotton species were used in analysis.

### 4.8. Promoter Regions Analysis of LACS Genes

For the identification of cis-elements in *LACS* genes, we got an up-stream sequence from CottonFGD of 2 kb of our genes from the translation start site. Subsequently, the sequences obtained were searched in PlantCare database (http://bioinformatics.psb.ugent.be/webtools/plantcare, accessed on 30 May 2022) [52] and eventually various cis-elements for our sequences were predicted.

### 4.9. Plant Materials and Growth Conditions

Two upland cotton plants CCRI70-177 (High SOC) and CCRI70-037 (Low SOC) were planted in Anyang, Henan, China (36.06° N,114.49° E) in 2022. In Anyang, the cotton was grown in a single-row plot with 18–23 plants, with a plot length of 5 m and a row spacing of 0.38 m. Crop management practices were conducted by following local recommendations for cotton production. Ovules were sampled at five developmental stages at 5, 10, 20, 30 and 40 day post-anthesis (DPA). There were three biological replications in each stage, with 3–5 plants for each biological replication. These samples were immediately placed into liquid nitrogen and stored in a freezer at −80 °C until use.

### 4.10. RNA Extraction and Analysis of Real-Time Quantitative Reverse Transcription (qRT)-PCR

We have used FastPure Plant Total RNA Isolation kit for polysaccharides & polyphenolics-rich samples (Vazyme, Nanjing, China.) to extract total RNA from all samples of cotton plants which were further used to synthesize cDNA with TransScript^®^ All-in-One First-Strand cDNA Synthesis SuperMix for qRT-PCR (TransGen, Benjing, China) following manual instruction. Specific primers were used to amplify the fragments which were prepared using an online primer designing tool kit at primer3plus (https://www.primer3plus.com/index.html, accessed on 1 October 2022) as provided in Appendix A The ABI Prism 7500 system was used to perform qRT-PCR, with GhUBQ7 (DQ116441) as the internal control. Experiments were done with three independent biological replications, and the 2^−ΔΔCt^ method was used to calculate the relative expression of *GhLACS* genes.

## 5. Conclusions

In summary, 22, 20, 12 and 11 *LACS* gene family members in *G. hirsutum*, *G. barbadense*, *G. raimondii* and *G. arboreum* were identified, respectively. This study is the first to gain insight into *LACS* gene members in cotton. From the aspects of gene structure, evolution mode, expression type and gene function, the evolution process and role of *LACS* genes in the evolution of the cotton genome were analyzed. Cotton is not only the most important textile fiber crop, but also an important oil crop. Our research lays the foundation for discovering the genes related to lipid synthesis and degradation creating new cotton germplasm materials for high oil.

## Figures and Tables

**Figure 1 ijms-24-08530-f001:**
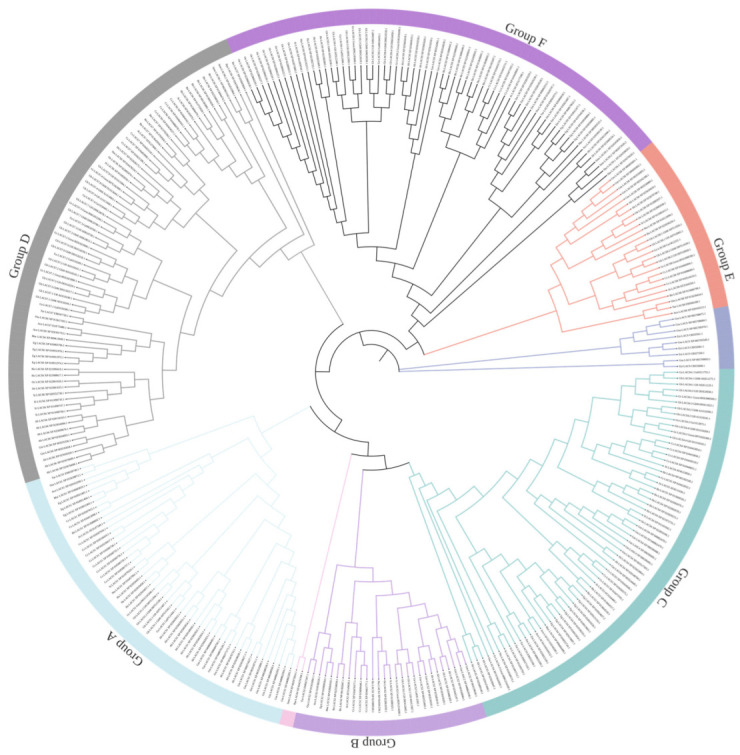
Evolutionary tree analysis (circle tree) and subgroup classifications of LACS proteins in cotton and other species.

**Figure 2 ijms-24-08530-f002:**
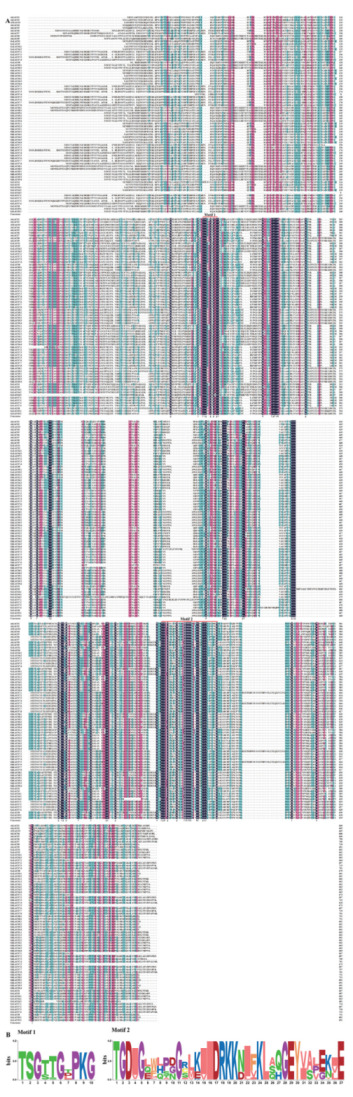
Multiple sequence alignment and conserved motif analyses of the GhLACS, GbLACS, GrLACS, GaLACS and AtLACS proteins. (**A**) Alignment of the amino acid sequences of the GhLACS, GbLACS, GrLACS, GaLACS and AtLACS proteins. Locations of the two conserved motifs are labeled with red lines. (**B**) Conservation of residues across the GhLACS, GbLACS, GrLACS, and GaLACS proteins are shown by the height of each letter. The bit scores indicate the information content for each conserved motif in the sequence.

**Figure 3 ijms-24-08530-f003:**
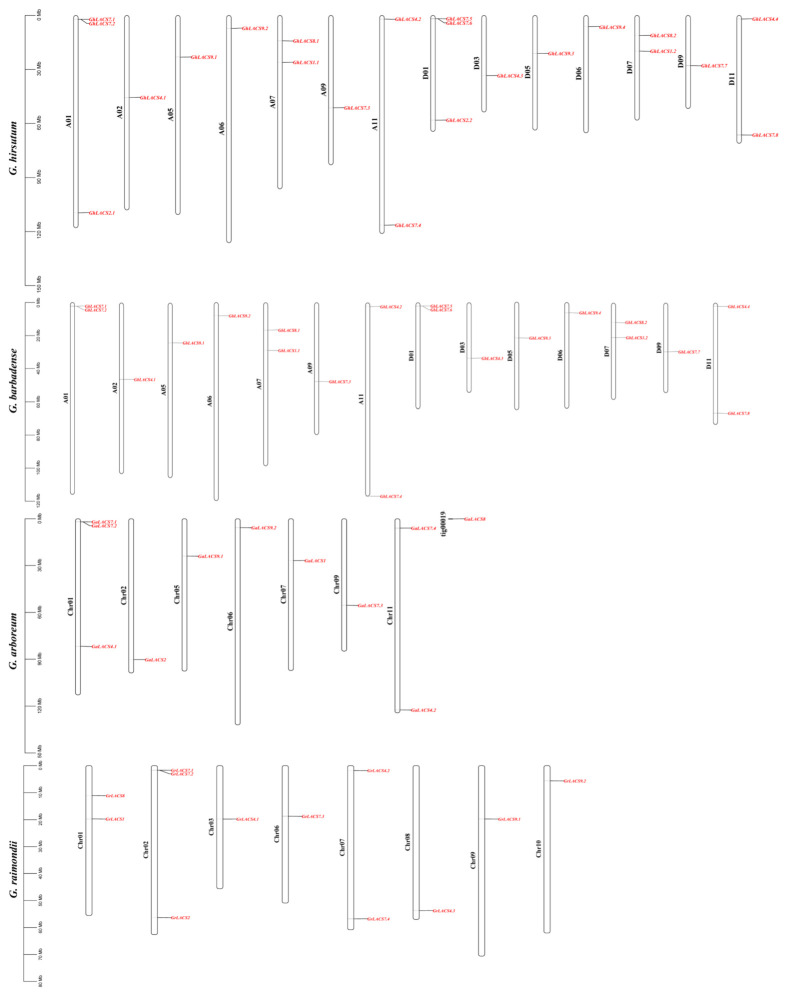
Chromosomal positions of *LACSs* from four cotton species with gene ids shown on the right side. The vertical bar on the left side represents the position of the gene and length of chromosome.

**Figure 4 ijms-24-08530-f004:**
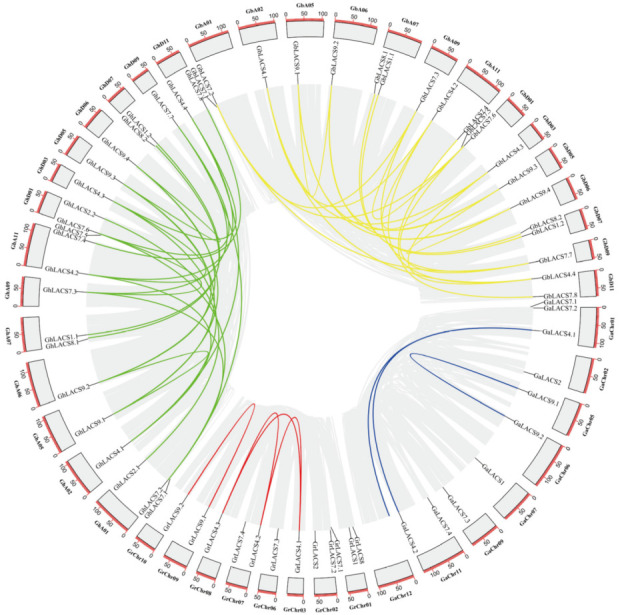
Gene duplication relationship among *LACSs* of four cotton species illustrated by various intra genomic syntenic regions represented with different colors. The scale on the circle is in Megabases. Each rectangle represents a chromosome.

**Figure 5 ijms-24-08530-f005:**
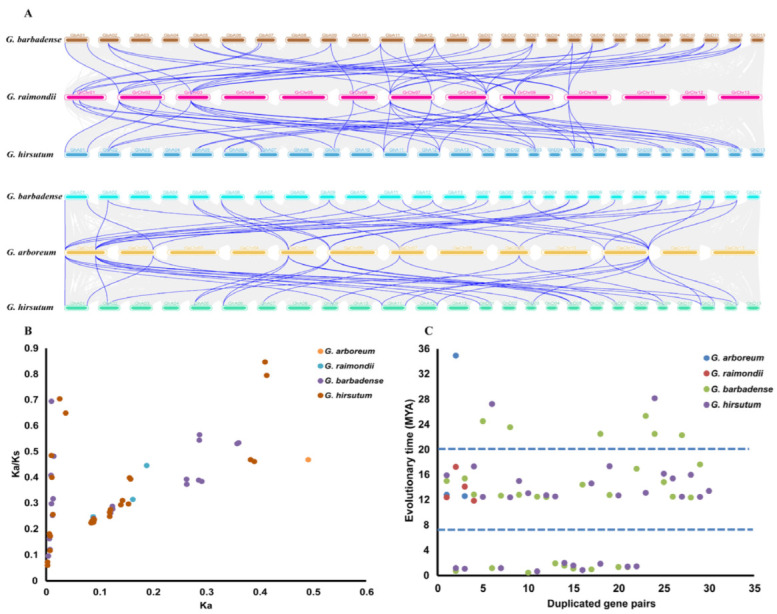
(**A**) Multiple collinearity analysis of *GhLACS* and *GbLACS* genes compared with their ancestor species through multiple synteny plots. Dense grey lines in the background revealed collinear blocks, blue lines represent syntenic *LACS* gene pair. (**B**) The Ka/Ks of 65 duplicated gene pairs in four cottons. (**C**) The evolutionary time of 65 duplicated genes pairs. Ka, the non-synonymous substitution rate; Ks, synonymous substitution rate; Mya, million years ago in four cottons.

**Figure 6 ijms-24-08530-f006:**
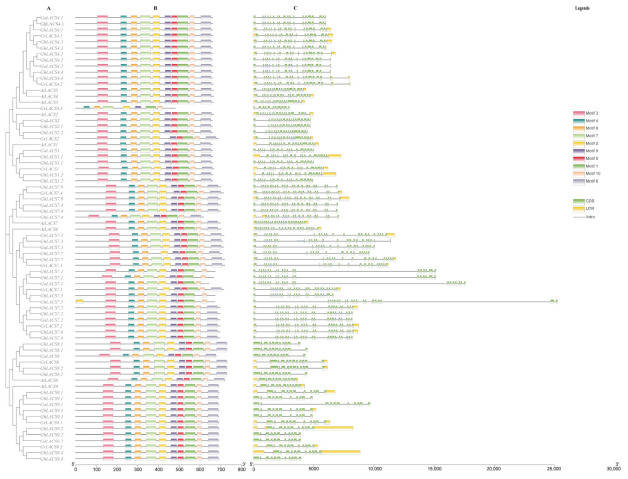
Phylogenetic relationship, exon-intron organizations, and motif analysis of LACS proteins. (**A**) represents the phylogenetic tree, (**B**) displays conserved motifs and (**C**) portrays the genomic structures. Legends: shows the legends for identification of various conserved motifs and exon/introns.

**Figure 7 ijms-24-08530-f007:**
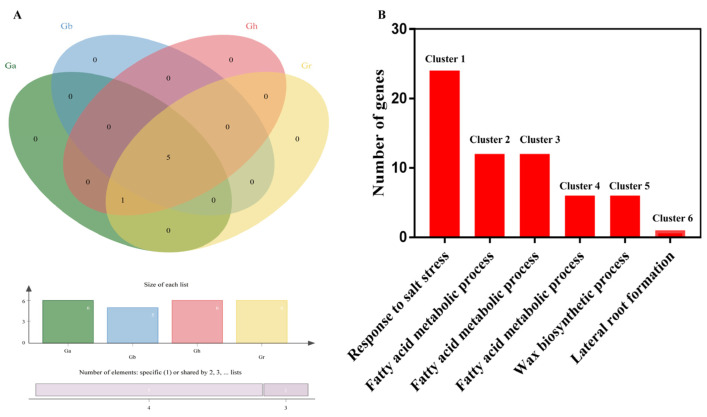
(**A**) Clustering of orthologous genes analysis between *G. arboreum*, *G. raimondii*, *G. barbadense* and *G. hirsutum* (**B**) gene annotation of orthologous gene clusters.

**Figure 8 ijms-24-08530-f008:**
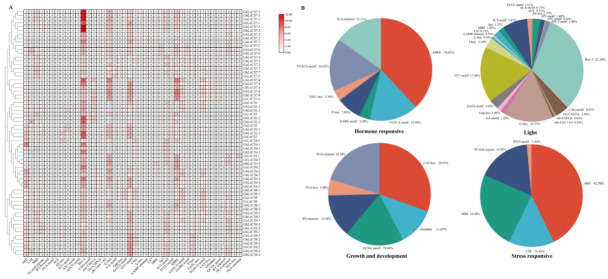
The cis-acting elements of LACS genes. (**A**) Numbers and gradient red colors indicate the number of cis-acting; (**B**) Pie charts show the proportion of different cis-acting elements in each category.

**Figure 9 ijms-24-08530-f009:**
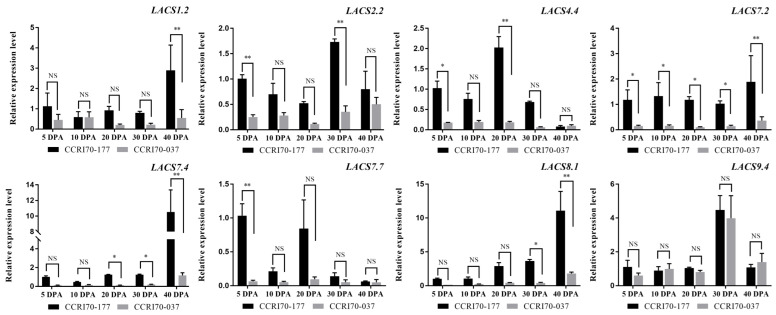
The relative expression profile of *GhLACS* genes in ovules of two upland cotton plants CCRI70-177 (High SOC) and CCRI70-037 (Low SOC). Each value represents the average of three biological repetitions. The error bars represented the SDs. An asterisk (*) indicates significance at *p* < 0.05, two asterisks (**) indicate significance at *p* < 0.01, NS indicates no significance.

**Figure 10 ijms-24-08530-f010:**
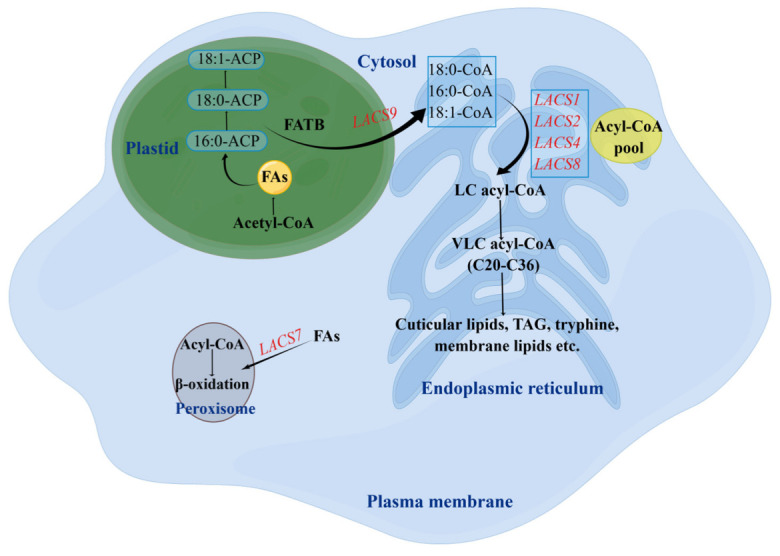
The possible roles of *LACSs* residing in different subcellular compartments. Abbreviations: FAs, fatty acid synthesis complex; FAE, fatty acid elongation complex; TAG, triacylglycerol; LC, long-chain; VLCA, very-long-chain.

**Table 1 ijms-24-08530-t001:** Comparison of chromosomes harboring number of *LACS* genes from different genomes and sub genomes of four cotton species (Ga, Gr, Gh, Gb), demonstrates possible gene loss and addition during evolution.

Chr. No	Ga	Gh-At	Gb-At	Gr	Gh-Dt	Gb-Dt	Total
Chr. 1	3	3	2	2	3	2	15
Chr. 2	1	1	1	3	0	0	6
Chr. 3	0	0	0	1	1	1	3
Chr. 4	0	0	0	0	0	0	0
Chr. 5	1	1	1	0	1	1	5
Chr. 6	1	1	1	1	1	1	6
Chr. 7	1	2	2	2	2	2	11
Chr. 8	0	0	0	1	0	0	1
Chr. 9	1	1	1	1	1	1	6
Chr. 10	0	0	0	1	0	0	1
Chr. 11	2	2	2	0	2	2	10
Chr. 12	0	0	0	0	0	0	0
Chr. 13	0	0	0	0	0	0	0
Scaffolds	1	0	0	0	0	0	1
Total	11	11	10	12	11	10	65

## Data Availability

The data presented in this study are available in the article and Appendix A.

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
