# Peer review of "Genome-Wide Analysis and Functional Characterization of *LACS* Gene Family Associated with Lipid Synthesis in Cotton (*Gossypium* spp.)"

_ijms, 2023, doi:10.3390/ijms24108530_

Round 1

Reviewer 1 Report

First of all, I would like to congratulate the authors of this article, as I think they have done an outstanding job.

The article “Genome-Wide Analysis and Functional Characterization of LACS Gene Family Associated with Lipid Synthesis in Cotton (Gossypium spp) describes the identification and characterization of the long-chain acyl-coenzyme A (CoA) synthetase (LACS) gene family in four different species of cotton. The study proposed LACS gene models and their potential role in lipid metabolism and demonstrated their potential for modulating TAG synthesis in cotton and genetic engineering of cottonseed oil. The article is well aligned with the aims and scope of International Journal of Molecular Sciences.

In reference to the use of English, the style is appropriate and understandable. The number of tests done and results obtained is solid and adequate. The information exposed in Materials and Methods is detailed enough.

In reference to specific changes, I suggest the following:

- In my opinion, the main element to improve in the work is the quality of the figures. Figures 1, 2, 3, 4, 5, 6 and 8 have many unreadable elements. I understand that the amount of information to be displayed is a lot, but the figures should have a higher resolution to be able to distinguish the text if it is enlarged.

- Line 34: The F in "fatty" should not be capitalized.

- Line 233: Reference 30 is after a full stop. Move it before.

- Foot of Figure 7: Species names should be in italics.

- Line 344: Species names should be in italics.

Reviewer 2 Report

The authors' article is based on a very interesting idea. Indeed, all cotton breeding has been aimed either at obtaining varieties with long, high-quality fibers or with highly oil-bearing seeds. And as a rule these were mutually exclusive. Indeed, it would be interesting to obtain a variety with high quality traits for both options. The authors did a huge bioinformatic pool of research to analyze lipid synthesis genes. In general, this part of the manuscript even looks a bit bloated, because if the abstract is already a bit forgotten, at some point one thinks that the whole article will end up in bioinformatics alone. Perhaps this part of the manuscript needs to be shortened a bit. But this is a matter of taste, as they say. In my opinion the manuscript is very well written and performed at a high methodological level. The data obtained by the author look reliable and can be trusted.

The only comment I have is about the diagram in Fig. 10. The picture of compartments of the cell is no good, I just "hung up" on it and for a long time I couldn't understand where and what? The plastid is in no way a bubble, and as a rule, lipid synthesis is most active in chloroplasts, which as usual are depicted in green and are bimembranous. The endoplasmic reticulum on the contrary is unimembranous, no need to frame it, it immediately begins to look like a mitochondrion with cristae. The peroxisome compared to the chloroplast is usually always smaller and has a single membrane. Please make the diagram more visually appealing.

Reviewer 3 Report

The authors conducted a study on the LACS gene family for four Gossypium species to determine the potential key points for synthetic cottonseed oil. The aim of this study is clear and the manuscript is of interest to me. Overall, I do not have major concerns with this manuscript, but I do have some technical and minor suggestions for improvement. 

Firstly, the authors used the neighbor-joining method to reconstruct the demographic patterns of the LACS gene family. However, the theory of the neighbor-joining method is based on genetic distance, whereas mutations for a gene family should be related to individual events for each mutation. Here, methods such as ML or MP would be better suited for using independent mutations as characteristics. Additionally, the substitution model needs to be tested before conducting the phylogenetic analysis. While the structure of the phylogenetic tree should remain the same, the meaning of the method is different. 

Secondly, for the Ka/Ks analysis, I suggest that the authors separate the orthologous and paralogous genes and recheck the analysis. Furthermore, the evolutionary analysis for selection can be applied by site, branch, or site/branch models. In the current analysis, it shows purifying selection. However, this is only an overall view. If the authors focus on different sites or branches, it can reveal further information on selection.

The English is find, but can be polished by Native English speaker. 
